# Comparison of sequence- and structure-based antibody clustering approaches on simulated repertoire sequencing data

Katharina Waury [iD] [1,2], Stefan Lelieveld [3], Sanne Abeln [iD] [1,2]*, Henk-Jan van den Ham [iD] [3,4]*

1 Department of Computer Science, Vrije Universiteit Amsterdam, Amsterdam, The Netherlands, 2 AI Technology For Life, Department of Information and Computing Science, and Department of Biology, Utrecht University, Utrecht, The Netherlands, 3 ENPICOM B.V., 's-Hertogenbosch, The Netherlands, 4 The Hyve B.V., Utrecht, The Netherlands

* s.abeln@uu.nl (SA); henk-jan@thehyve.nl (HJvdH)

**Data availability statement:** All data and code relevant to this study are available at https://github.com/kathiwaury/clustering-comparison.

## Abstract

Repertoire sequencing allows us to investigate the antibody-mediated immune response. The clustering of sequences is a crucial step in the data analysis pipeline, aiding in the identification of functionally related antibodies. The conventional clustering approach of clonotyping relies on sequence information, particularly CDRH3 sequence identity and V/J gene usage, to group sequences into clonotypes. It has been suggested that the limitations of sequence-based approaches to identify sequence-dissimilar but functionally converged antibodies can be overcome by using structure information to group antibodies. Recent advances have made structure-based methods feasible on a repertoire level. However, so far, their performance has only been evaluated on single-antigen sets of antibodies. A comprehensive comparison of the benefits and limitations of structure-based tools on realistic and diverse repertoire data is missing. Here, we aim to explore the promise of structure-based clustering algorithms to replace or augment the standard sequence-based approach, specifically by identifying low-sequence identity groups. Two methods, SAAB+ and SPACE2, are evaluated against clonotyping. We curated a dataset of well-annotated pairs of antibodies that show high overlap in epitope residues and thus bind the same region within their respective antigen. This set of antibodies was introduced into a simulated repertoire to compare the performance of clustering approaches on a diverse antibody set. Our analysis reveals that structure-based methods do group more antibodies together compared to clonotyping. However, it also highlights the limitations associated with the need for same-length CDR regions by SPACE2. This work thoroughly compares the utility of different clustering methods and provides insights into what further steps are required to effectively use antibody structural information to group immune repertoire data.

**Funding:** K.W. and S.A. received funding from the European Union's Horizon 2020 research and innovation program under the Marie Sklodowska-Curie grant agreement no. 860197, the MIRIADE project (https://miriade.eu/). The funding agency did not play any role in the study design, data collection and analysis, decision to publish, or preparation of the manuscript.

**Competing interests:** I have read the journal's policy and the authors of this manuscript have the following competing interests: S.L. declares that he is an employee of ENPICOM B.V.. H.J.v.d.H. declares that he is an employee of The Hyve B.V. and a former employee of ENPICOM B.V.. S.A. and K.W. state that they have no competing interests.

## Author summary

Understanding our adaptive immune system response is crucial for developing vaccines and therapies. Grouping antibodies based on their function helps us understand the diversity of immune cells and allows us to find valuable therapeutic antibody candidates faster. In this study, we compare different methods for grouping antibodies, i.e., clustering approaches. The traditional method, known as clonotyping, groups antibodies based on sequence similarity and identical gene usage. However, this approach may miss functionally related antibodies that do not have a similar sequence and derive from differing genes. We explore the potential of structure-based clustering methods, which have become feasible on a large scale recently, to identify such antibodies. We compare two structure-based methods, SAAB+ and SPACE2, against clonotyping by creating a dataset of antibodies known to bind the same region of their target and integrating them into a simulated but diverse set of antibodies, called a repertoire. Our findings show that structure-based methods can form larger groups of related antibodies, but also face challenges such as missing structure templates and the requirement for same-length CDR regions. We highlight the promise and the current hurdles in using structural information to enhance antibody repertoire analysis.

## Introduction

High-throughput repertoire sequencing has emerged as a fundamental tool to investigate and understand the human immune system [1]. B-cell receptor (BCR) sequencing data, in particular, provides insights into the antibody-mediated adaptive immune response. Investigating these processes is crucial to advance our understanding of modern health challenges including autoimmunity and vaccine development [2,3]. An important analysis step in processing BCR sequencing data involves the grouping of the sequences into clonotypes (or clonal groups), based on the same clonal lineage, which is hereinafter referred to as "clonotyping" [4,5]. Importantly, clonotyping facilitates the identification of antibodies that A) genetically share a common originator cell, and B) exhibit a highly similar function, i.e., recognize the same antigen. The shared origin of two antibodies is typically established through usage of identical V and J germline genes. The functional similarity is determined by a high sequence identity within the complementarity-determining region (CDR) 3 of the heavy chain (CDRH3). Commonly, a CDRH3 sequence identity of ≥80% at the amino acid level is used to group antibodies as one clonotype together, although a higher cutoff can be applied if warranted [5]. The emphasis on the CDRH3 region during clonotyping is due to its hypervariability and its strong involvement in determining antibody binding [6,7]. Identical CDRH3 length may be an additional criterion for grouping antibodies [4]. Clonotyping is an entirely sequence-based approach and the current standard for clustering of B-cell repertoires [8].

For the identification of the full set of same-epitope-binding antibodies within a repertoire, clonotyping has its limitations. Several studies have reported antibodies that bind the same epitope despite differing clonal lineage [9–11]. Thus, the repertoire may contain antibodies that have "functionally converged" despite their genetically differing origins [8]. Identification of antibodies with the same binding specificity, also called "paratyping", is of high interest to discover and engineer novel therapeutic antibody candidates [12]. Clonotyping would inadvertently overlook some of them because of the reliance on sequence similarity to define groups.

As structure is more closely linked to function than sequence in folded proteins [13], using structural antibody information has received increased interest for the analysis of repertoire sequencing data [14–16]. The high workload and time-intensive nature of experimental structure determination methods, such as X-ray crystallography, render them impractical for studies comprised of more than a handful of antibodies, and can thus not be applied to BCR sequencing datasets [15]. Computational modeling of antibody structures, on the other hand, is a feasible approach to annotate entire repertoires. Importantly, the antibody field has benefited from the enormous advances in protein structure prediction [17] and protein embeddings [18,19]. By refining and extending upon these innovations, novel antibody structure prediction models have been developed, specifically tailored to address the intricate challenges of immune cells [20,21].

In one of the first studies incorporating large-scale antibody structure modeling, de Kosky et al. [22] included thousands of RosettaAntibody-predicted models in their analysis of the systematic repertoire changes elicited by an immune response. More recently, structure-based clustering of BCR data has been pursued, promising to overcome the limitations of clonotyping, and to group together antibodies with a higher genetic diversity [8,23,24]. Structure-based approaches use root mean square deviation (RMSD) to quantify similarity between two structures. A lower RMSD indicates higher structure similarity, thus, two structures are grouped together if their RMSD is below a set threshold.

One early implementation, SAAB+, was developed to investigate structural diversity in repertoires. The tool depends on homology modeling to assign antibodies to structural clusters by aligning the backbone atoms of the CDR3 regions and calculating the RMSD [25]. While limited by the availability of experimentally solved antibody structures, the method annotated whole repertoires swiftly and revealed differences in structural diversity across B-cell types [25].

More recently, SPACE2 was developed. Here, antibodies are aligned on their framework regions and clustered based on the structural similarity of the backbone residues of the CDR loops. SPACE2 requires numbered antibody structures as input such as those generated by the *ab initio* antibody structure predictor ImmuneBuilder [20,24]. The authors of SPACE2 compared their work to clonotyping, highlighting the potential of SPACE2 to create more consistent multiple-occupancy clusters, i.e., clusters that contain two or more unique antibody sequences with the same epitope binding region [24].

These studies indicate that structure-based grouping can indeed identify additional functionally converged antibody pairs in repertoire data. While encouraging, it is important to note that SPACE2 and other recent approaches have been trained and tested on datasets enriched for a specific antigen [12,24,26]. Additionally, while the first benchmark of sequence- and these novel structure-based clustering algorithms was recently published, the study involved the same data used to train SPACE2, i.e., a set of antibodies against the receptor-binding domain (RBD) of Severe acute respiratory syndrome coronavirus 2 (SARS-CoV-2) [27]. Note that this dataset provided epitope groups based on epitope-binning experiments rather than residue-level epitope annotations [28].

Testing of these methods on data that more accurately represents the diverse repertoire data they are meant to analyze is still missing. Such an investigation is imperative to comprehensively assess the utility of structure-based clustering in the broader context of BCR repertoire analysis. Most importantly, it is vital to confirm if these novel types of approaches are able to identify functionally converged antibodies while also grouping clonally related antibodies.

Here, we aim to fill this gap by comparing the performance of sequence-based clonotyping with novel structure-based approaches applied to simulated repertoire sequencing data.

The included methods, IGX-Cluster (a pipeline for clonotyping), SAAB+ and SPACE2, are summarized and compared in Table 1. To obtain a set of antibody pairs for which the ground truth regarding functionality is known, we curated a dataset of functionally similar antibodies, both with and without high CDRH3 sequence similarity. Subsequently, these well-annotated antibody pairs were introduced into a simulated repertoire dataset. This approach allows to assess the performance on a diverse and non-enriched set of BCR sequences. In addition, a baseline for "random" clustering rates can be established across all included methods. The further focus of analysis revolves around the capabilities of the different approaches to accurately identify and group functionally related antibody pairs together, both clonally related and evolutionary converged ones. Thereby, this study provides insights into their effectiveness in capturing relevant structural and functional relationships within a diverse antibody repertoire.

## Results

### Functional convergence is confirmed in antibodies binding to well-studied antigens

To create a set of functionally similar antibody pairs, the Immune Epitope Databases (IEDB) [29] was searched for epitopes located on the same protein antigen. If the overlap of the residues of two epitopes was ≥75% as defined by their Jaccard index, the respective antibodies were defined as functionally similar and retained. This overlap cutoff allowed the inclusion of a sufficient number of antibody pairs. The epitope information of these functionally similar antibody pairs was cross-referenced with SAbDab [30] and the Protein Data Bank (PDB) [31] to obtain relevant annotations. Combining the annotations from all three resources led to a final dataset of 213 antibody pairs comprised of 54 unique antibodies (Fig 1). For all antibodies in this dataset, the antibody amino acid sequence, the PDB antibody structure, and the residues within the antigen binding region are known. For every pair of antibodies, the Levenshtein distance was calculated to assess the sequence identity of the CDRH3 regions. Complete information for each pair, including the IEDB epitope ID and the PDB ID of the antibody structure, as well as its cluster assignment can be found in the supplement (S1 Table).

The limited number of antibody pairs is unsurprising considering it requires the thorough investigation of multiple antibodies for the same antigen. While a multitude of more antibody pairs with high functional similarity was identified using the IEDB, only a small subset had a solved PDB structure available for both antibodies. The majority of antibodies bind to SARS-CoV-2 proteins, other antigens originate from the influenza and human

**Table 1. BCR repertoire clustering approaches.**

| Cluster approach | Method | Input | Cluster properties | Limitations | Tuneable parameters | Ref. |
|---|---|---|---|---|---|---|
| Clonotyping | IGX-Cluster | Antibody nucleotide sequence | V gene call, J gene call, CDRH3 sequence identity, CDRH3 sequence length | Unable to detect functionally converged antibodies; potential errors during reference gene assignment | CDRH3 sequence identity cutoff | [5] |
| Structure-based clustering | SAAB+ | Antibody heavy chain amino acid sequence | RMSD of CDRH3 loop (of most similar PDB template) | Requires a similar CDRH3 loop PDB structure for homology modeling; does not provide a cluster for every antibody | - | [25] |
| | SPACE2 | Predicted antibody structure | RMSD of CDR region backbone atoms | Requires all CDR region to have identical length | RMSD cutoff | [24] |

CDR - complementarity determining region, RMSD - root mean square deviation.

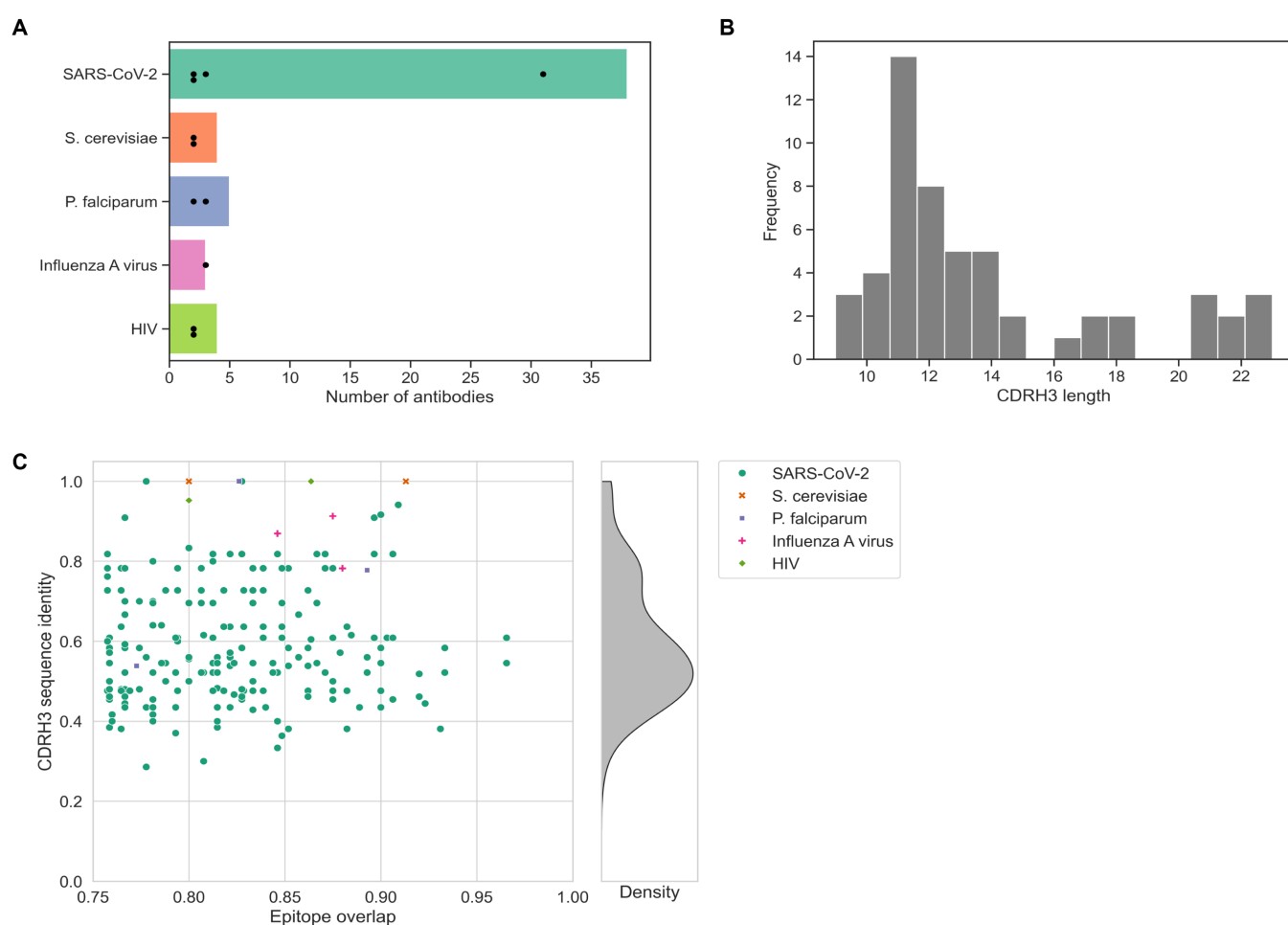

**Fig 1. Antibody pair dataset characteristics.** A dataset of antibody pairs with similar function, i.e., highly overlapping antigen binding regions, was created and annotated. The final set contains 213 antibody pairs comprised of 54 unique antibodies. A: The included antibodies bind to one of five protein antigens of well researched species. Bars indicated the number of antibodies associated with each species, the dots indicate the sizes of the antibody pair clusters into which the antibodies group. The majority of antibodies bind to SARS-CoV-2 derived antigens. B: The CDRH3 amino acid sequence length of the included antibodies ranges from 9 to 23, the most common length is 11 amino acids. The diversity of CDRH3 sequence length is in agreement with previous observations. C: A scatter plot shows the CDRH3 amino acid sequence identity and epitope overlap of each antibody pair. Color and marker style indicate the antigen species. Only antibody pairings with an epitope overlap of ≥0.75 are included. A kernel density estimate plot indicates the distribution of CDRH3 sequence identity across the dataset.

immunodeficiency virus (HIV) (Fig 1A). All antibodies bind well-studied antigens that are of high interest because of their importance within infection or autoimmunity research. Functionally grouping these antibodies, i.e., based on their epitope overlap, results in one large cluster comprised of 31 SARS-CoV-2 antibodies and ten small clusters comprised of two or three antibodies (Fig 1A).

The range of observed CDRH3 lengths in this dataset is large (Fig 1B) but is in agreement with previous studies on CDR length diversity [32]. The epitope overlap between non-identical antibodies ranges from 75.76% to 96.55% (Fig 1C). Importantly, the dataset contains antibody pairs both with high and low sequence identity in the essential CDRH3 region (Fig 1C) confirming that functional convergence can lead to low sequence similarity antibodies that bind the same antigen region.

## Backtranslated antibody sequences are indistinguishable from simulated BCR repertoire background

To compare the different clustering strategies within a realistic repertoire sequencing setup, we wished to introduce the annotated sequences of our functionally similar antibody pair dataset into a larger BCR repertoire simulated using the immuneSIM package [33]. To be able to add the curated antibodies, their amino acid sequences were backtranslated to nucleotide sequences guided by the nucleotide sequences of the most likely V and J reference genes. The somatic hypermutation (SHM) rate of the simulated repertoire was set to match the SHM rates observed in the antibody pair dataset. In brief, a low and a high SHM rate was inferred from the antibody pair sequences; three repertoires of no, low and high SHM rate were then simulated and combined. More details on the processes of sequence backtranslation and SHM rate calculation can be found in Methods. The reconstructed antibody pair nucleotide sequences were then introduced into the simulated repertoire resulting in our full repertoire dataset of 10.490 unique sequences. Note that functional annotations are only available for the 54 curated antibodies, not the simulated antibodies.

To ensure that the antibody pair sequences do not distinctly differ from the simulated background sequences because of any bias introduced during the sequence backtranslation, we applied principal component analysis (PCA) for dimensionality reduction and visualization. The PCA was fitted on relevant sequence descriptors, including the number of mutations and receptor length, and the first and second principal component were visualized (S1 Fig). As the curated and simulated antibodies overlap strongly with each other, their sequence properties are not systematically different. Thus, the downstream analyses should not be affected by any bias within the sequences.

## Clustering approaches correctly identify small and partly overlapping subsets of functionally similar antibody pairs

To be able to compare clustering strategies, the full repertoire sequencing dataset was processed using the IGX platform[34]. Antibodies were annotated from the sequencing data, with the closest V and J reference gene, as well as somatic hypermutations being identified. The repertoire sequences were clustered using the different methods described in the introduction: sequence-based clonotyping using IGX-Cluster, structure-based clustering using SAAB+ and SPACE2, respectively.

The performance of the tools, in regards to correctly grouping the annotated functionally similar antibody pairs, is summarized in Table 2 and visualized in Fig 2. Across the methods, all and solely multiple-occupancy clusters, i.e., clusters with two or more differing antibody sequences, are compared. For details on the metrics, see Methods. IGX-Cluster assigns all unique sequences to clusters. SPACE2 assigns all but one antibody as for one sequence no structure could be produced by ImmuneBuilder. SAAB+ clusters 87.48% of sequences; for the

**Table 2. Results of simulated repertoire clustering.**

| Method | Number of clustered antibodies | Number of clusters | Mean cluster size | Number of multiple-occupancy clusters | Mean multiple-occupancy cluster size | Clustered antibody pairs | Clustered low-sequence similarity antibody pairs |
|---|---|---|---|---|---|---|---|
| IGX-Cluster | 10490 | 10197 | 1.03 | 227 | 2.29 | 24 (11.27%) | 8 |
| SAAB+ | 9100 | 973 | 9.35 | 763 | 11.65 | 16 (7.51%) | 5 |
| SPACE2 | 10489 | 8031 | 1.31 | 1452 | 2.69 | 14 (6.57%) | 5 |

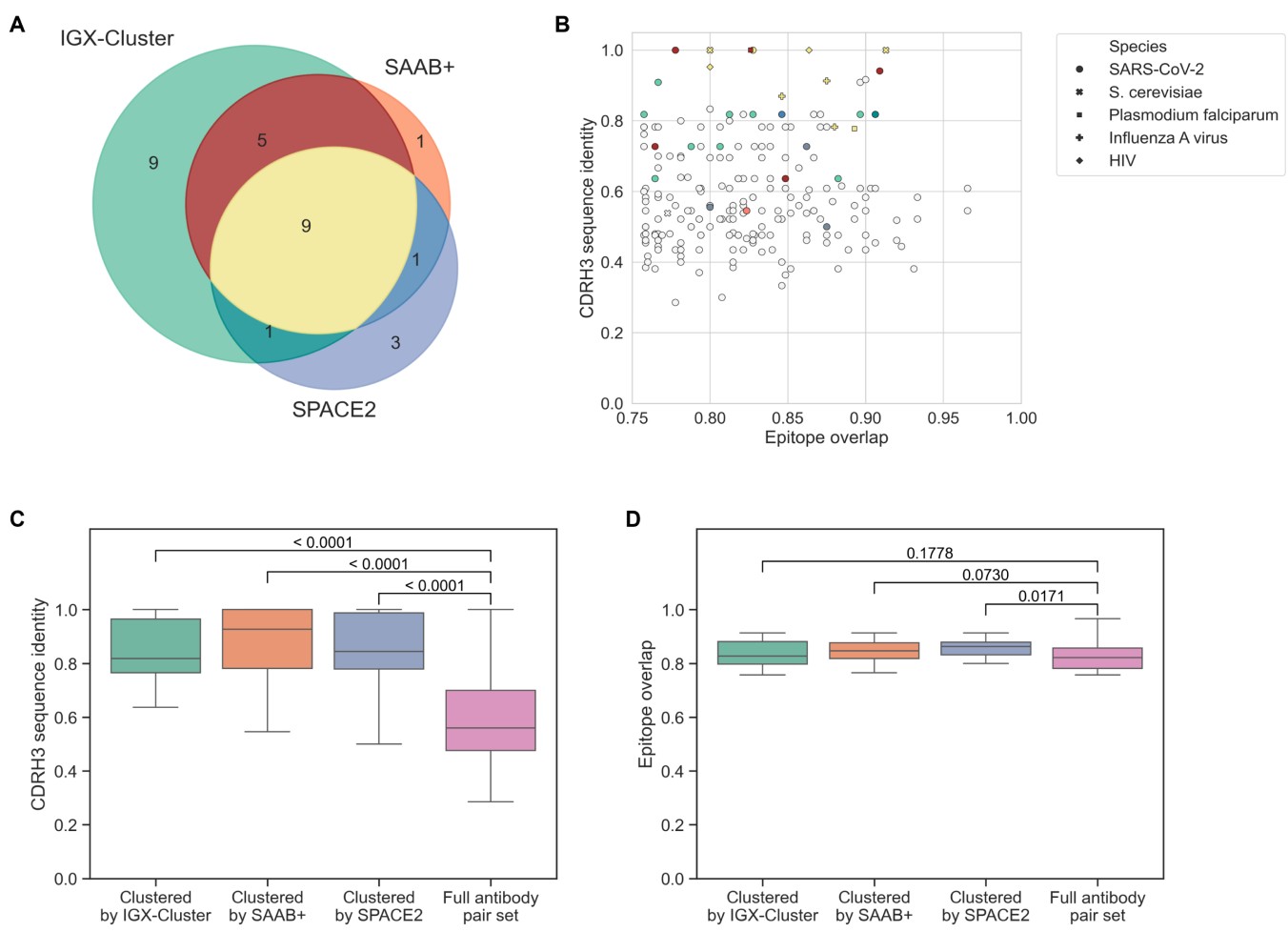

**Fig 2. Performance comparison of different clustering strategies.** The three approaches, clonotyping, SAAB+ and SPACE2, were applied to the repertoire. Their performance on the annotated set of 213 functionally similar antibody pairs was evaluated. A: Euler diagram showing the overlap of correctly clustered antibody pairs between methods regarding. B: A scatter plot shows the CDRH3 sequence identity and epitope overlap of each antibody pair. Marker style indicates the antigen species. Color indicates which methods correctly grouped each antibody pair (see Euler diagram for color code). The majority of antibody pairs have not been identified together by any methods (gray, 184 antibody pairs). C: All three strategies cluster antibody pairs with a significantly higher CDRH3 sequence identity compared to the full antibody pair set. D: The epitope overlap of clustered antibody pairs is similar between the methods, albeit SPACE2 identified antibody pairs with a slightly higher epitope overlap compared to the full set. Statistical significance was tested using the Wilcoxon rank-sum test.

remaining no CDRH3 template was available in the PDB. However, for all sequences of the antibody pair set a cluster was assigned.

The majority of antibody pairings was not identified by any of the three approaches (184 pairs, 86.38%). The most antibody pairs were correctly grouped together by the sequence-based clonotyping approach (24 pairs, 11.27%). SAAB+ and SPACE2 identified 16 (7.51%) and 14 pairs (6.57%), respectively. Thus, all methods show a low sensitivity for our set of antibody pairs. On the other hand, all three methods show high precision as no two antibodies were incorrectly assigned to the same cluster. Comparing the overlap of identified antibody pairs, each clustering approach grouped a partly different set of antibody pairs (Fig 2A and 2B). This finding of differing groupings is in line with the recent benchmark of

clustering strategies tested on SARS-CoV-2 antibodies [27]. Of the 13 antibody pairs only clustered together by one of the methods, the majority was identified by clonotyping.

We further investigated if the structure-based approaches deliver on their promise of grouping antibodies together with a lower sequence dissimilarity as these tools are not confined by the requirement of identical reference genes or a minimum sequence identity within the CDRH3 region. Comparing the normalized CDRH3 distance of the antibody pairs correctly clustered by each method shows no significant differences between the standard approach of clonotyping and structure-based tools (Fig 2C). All compared methods group antibody pairs with a significantly higher sequence identity compared to the full antibody set. Note, however, that both SAAB+ and SPACE2, correctly identify a limited number of antibody pairs with low sequence identity in the CDRH3 region (Fig 2B).

Comparing the epitope overlap of the antibody pairs identified by each method shows no strong difference between methods and the full antibody pair dataset (Fig 2D). This observation indicates that our chosen epitope overlap cutoff of ≥75% ensures that all included antibody pairs are functionally similar enough to be grouped together.

As no clustering method was able to correctly group a large fraction of the annotated antibody pairs, we further investigated the potential limitations of each workflow in detail.

## Clonotyping is limited but highly accurate

Standard clonotyping is restricted two-fold: for sequences to be assigned to one clonotype, they need to be designated the same V and J gene and exhibit a CDRH3 sequence identity above a certain threshold. If considering these two requirements, clonotyping correctly grouped all possible antibody pairs of our curated dataset correctly, i.e., all antibody pairs with identical V and J gene assignment and a CDRH3 sequence identity ≥80% (Fig 3A). Interestingly, clonotyping clustered antibody pairs below the threshold as well because of its hierarchical clustering approach. In fact, clonotyping identified more low-sequence similarity antibody pairs than SAAB+ and SPACE2 from our dataset (Table 2). These antibody pairs of lower sequence identity are part of the large cluster of functionally related SARS-CoV-2 antibodies (Fig 1A). Within this cluster several antibodies form multiple antibody pairs allowing antibody pairs below the sequence identity threshold to be clustered together. To illustrate, antibody pairs A+B and A+C have a sequence similarity of 0.9 and are correctly clustered together by clonotyping. In this case, even if the antibody pair B+C has a low sequence similarity, e.g., of 0.7, this functional pair is still correctly clustered together because of their respective pairing with antibody A.

The number of antibodies clustered together by clonotyping in the full repertoire is very low (Table 2). This is expected as the simulated repertoire was naive, i.e., not antigen-experienced thus no clonal expansion was expected. The only enrichment of functionally similar antibodies thus comes from the included antibody pair dataset. If larger clusters are desired, one strategy is to only use the V gene for partitioning and ignore the J gene assignment. This less stringent criterion would lead to larger cluster sizes in the repertoire (Fig 3C). Examining only our annotated antibody pair dataset shows that the effect of abandoning the J gene requirement is in this case limited as only two additional antibody pairs without the J gene match show a CDRH3 sequence identity ≥80% (Fig 3A).

## Same CDR length requirement strongly limits clustering by SPACE2

Structure-based clustering methods are not constrained by sequence identity of the CDRH3 region. Hence, we expected more antibody pairs to be correctly identified by SPACE2 compared to clonotyping as our dataset contained many low sequence similarity antibody

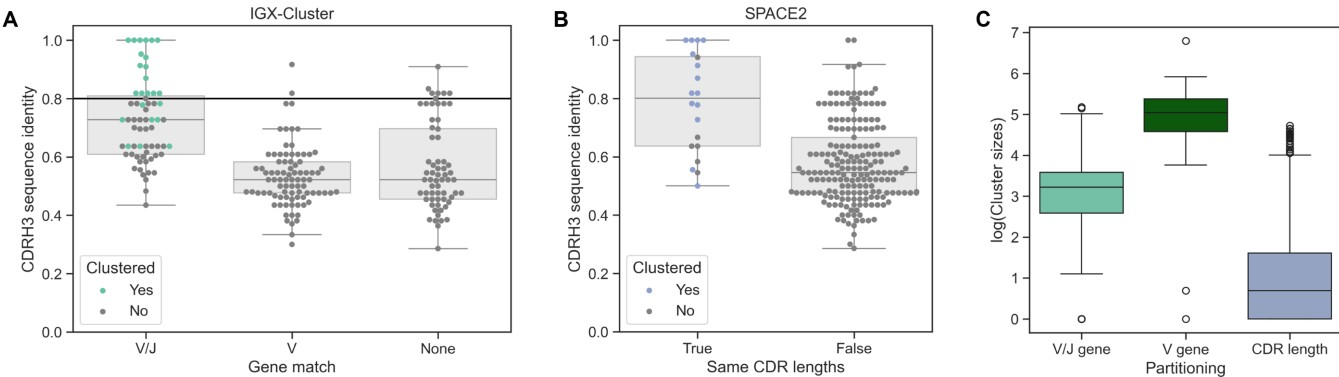

**Fig 3. Limitations of clustering approaches.** Both IGX-Cluster and SPACE2 partition the antibodies, before clustering based on CDRH3 sequence identity or CDR RMSD, respectively. Swarm and box plots show the distribution of antibody pairs across the partition strategies. Colored dots indicate the correctly identified antibody pairs for each method. A: Clonotyping partitions antibodies based on matching V and J genes. Antibodies with identical V and J gene usage have a higher sequence identity than antibodies with identical gene usage in only the V or none of the genes. Partitioning based on only the V gene can improve the coverage slightly but is limited by the low sequence identity between these antibodies. B: SPACE2 partitions based on same length in all six CDR regions. The majority of antibody pairs (193, 90.61%) do not meet this requirement. Of the antibody pairs with same CDR lengths, 70% (14 out of 20) are correctly grouped together. C: The natural logarithm of cluster sizes across the full repertoire dataset indicates how stringent the different partitioning strategies applied by either IGX-Cluster (dark and light green) or SPACE2 (blue) are. The criterion of same CDR region lengths is the most stringent, while requiring solely the same V gene is the least stringent and leads to the largest cluster sizes.

comparisons. As clonotyping outperformed SPACE2, we were interested to understand the limitations of the SPACE2 approach. Importantly, while SPACE2 clusters using the RMSD of the antibodies' predicted CDR loops, it partitions the dataset based on the length of all six CDR regions beforehand [24]. Comparison of CDR lengths for each antibody pair shows that the same CDR length requirement is strongly affecting the capabilities of SPACE2 to identify functionally similar antibodies in our dataset. The vast majority of antibody pairs (90.61%) show a difference in length in at least one of the CDR regions (Fig 3B). Crucially, the same CDR length criterion allows SPACE2 to group only a subset of higher CDRH3 sequence identity antibodies (Fig 3B). This finding explains why SPACE2 is grouping almost entirely high sequence identity pairs despite the structure-based strategy (Fig 2C). Of the antibody pairs with same CDR lengths 70% (14 out of 20) are correctly clustered together using the default RMSD cutoff of 1.25Å.

Examination of the full repertoire shows similar restrictions. The SPACE2-based partitioning leads to the lowest cluster size within the full repertoire suggesting that the CDR length criterion is the most restrictive strategy (Fig 3C). Note that partitioning based solely on the CDR lengths already creates 1891 distinct clusters in the repertoire of which 808 (42.73%) contain only a single unique sequence.

## Less stringent clustering strategies provide a better trade-off between sensitivity and precision

While SAAB+ grouped more antibody pairs than SPACE2, the method produced clearly larger clusters across the full repertoire dataset (Table 2). Thus, we wished to establish the random clustering rate of each method to take into account correct cluster assignment of an antibody pair by chance. As expected, the larger average cluster size of SAAB+ leads to a high random clustering rate compared to clonotyping and SPACE2 (Fig 4A). However, the likelihood of grouping the antibody pairs by chance is still low, as on average less than one

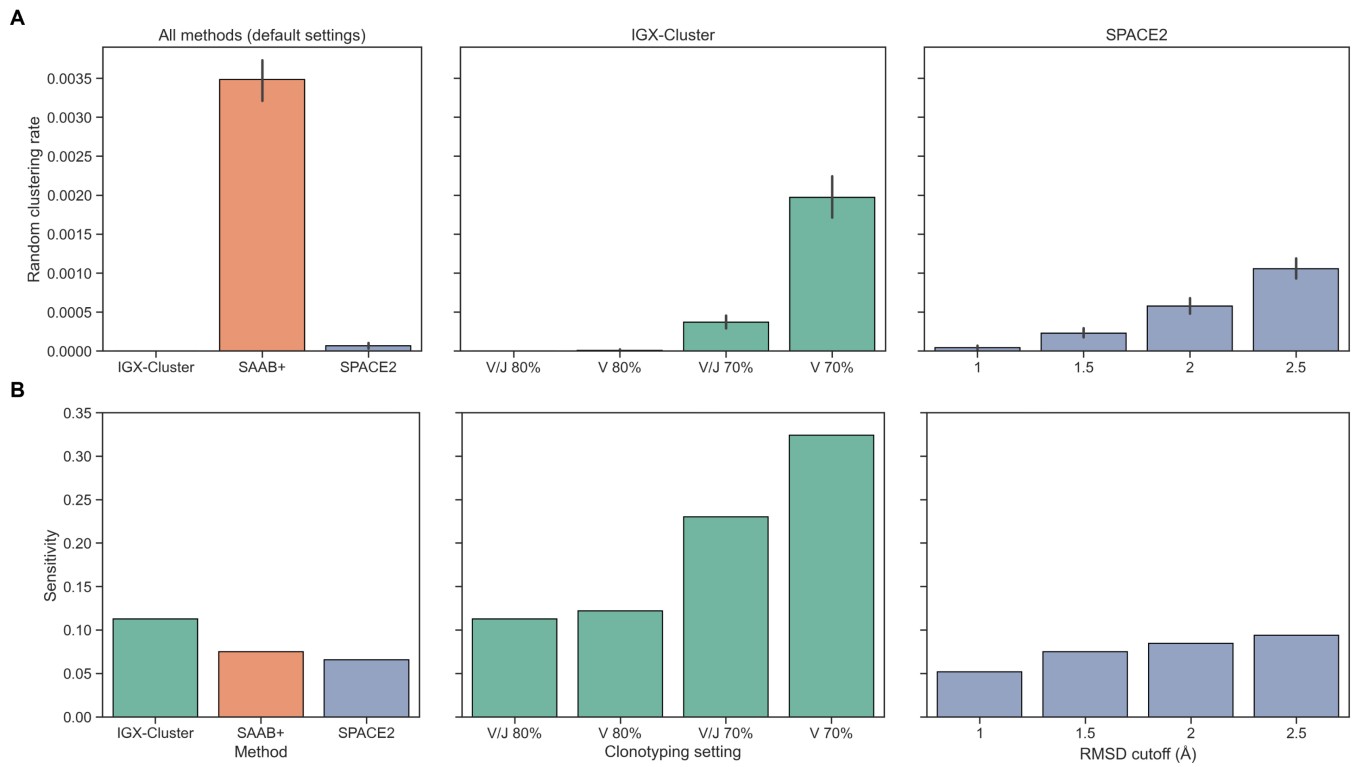

**Fig 4. Random clustering rates and sensitivity of clustering strategies across varying settings.** A: To calculate the random clustering rates, the cluster size distributions for the full repertoire created by each approach were gathered. The repertoire antibodies were then randomly assigned to clusters of the same sizes to infer how likely random assignment to the same cluster is for the antibody pairs. Low rates indicate a low probability of grouping functionally similar antibody pairs by chance. Lowering the clustering requirements in the IGX-Cluster and SPACE2 setup increases the average cluster sizes and thus the random clustering rate, but it remains below the rate of SAAB+. B: Comparing different clustering settings shows that sensitivity can be increased while the precision remains at 100%, i.e., the number of falsely grouped antibody pairs does not increase. The sensitivity increase of SPACE2 is limited by the CDR same length requirement.

antibody pair was randomly assigned to the same cluster when considering the SAAB+ cluster sizes.

None of the approaches grouped any erroneous antibody pairs, i.e., there are no false positives. Considering how precise all three strategies performed, we investigated what effect less stringent requirements have on the random clustering rate.

We reevaluated the clustering of the repertoire with differing clonotyping settings which either exclude the requirement of identical J gene usage or lower the minimum sequence identity of the CDRH3 region to 70%. As expected, these less stringent criteria lead to larger clusters and a higher random clustering rate (Fig 4A). However, a higher number of antibody pairs are correctly identified (Fig 4B), i.e., the sensitivity is significantly higher. Importantly, none of our curated antibody pairs are falsely clustered together, even in the least stringent clonotyping setting.

Similarly to IGX-Cluster, we examined the effect of relaxing the clustering requirements on the random clustering rate and the sensitivity of SPACE2. This translated to increasing the RMSD cutoff to group structures together. Higher RMSD cutoffs lead to larger clusters and a higher random clustering rate. However, even at an RMSD cutoff of 2.5Å, the rate is still lower compared to SAAB+, i.e., the SPACE2 strategy shows higher precision (Fig 4A). The increase in sensitivity is less significant compared to IGX-Cluster, as the same CDR length criterion

limits the number of detectable antibody pairings so severely. At the highest included cutoff of 2.5Å, SPACE2 correctly identifies all 20 antibody pairs with identical CDR lengths. A further increase of the RMSD cutoff could thus not improve sensitivity of the method any further. To establish whether SPACE2 clustering works specifically due to the structural information, we compared the SPACE2 algorithm to an approach only utilizing the CDR length criterion and the CDR length criterion in combination with a sequence identity cutoff of 80% in the CDRH3 region. Neglecting structural information leads to a higher random clustering rate with identical sensitivity (S2 Fig), suggesting that the structure-based clustering does improve results. Further, clustering based on CDR length matching and CDRH3 sequence identity performs slightly worse than the default SPACE2 algorithm, both regarding the random clustering rate and the sensitivity (S2 Fig). None of these approaches cluster wrong antibody pairs.

## Discussion

Incorporating structural information for antibody data analysis has attracted increased attention in recent years. Multiple approaches have been proposed for structure-based clustering of repertoire data to substitute for or augment clonotyping. As novel methods become available, a comparison of their performance with the previous standard technique is essential. Crucially, data used for evaluation should closely resemble the real-world datasets which these methods are intended to be applied to, a goal we aimed to achieve in this study.

Our study introduces a curated set of functionally similar antibody pairs. Importantly, this dataset contains antibodies recognizing various antigens and provides detailed annotations of sequence, structure, and binding region. Thus, this dataset serves as a valuable resource for further studies of methods aiming to identify functionally related antibodies. However, the small size of our annotated antibody set and the dominance of anti-SARS-CoV-2 antibodies, highlights that research is still severely limited by the scarcity of functionally annotated antibody data.

Further, we implemented the backtranslation of the antibody amino acid sequence to nucleotide sequence by considering available human reference gene sets. Importantly, this step allowed us to introduce the annotated antibodies into a simulated repertoire sequencing dataset which could then be used as the starting point of our cluster analysis workflow. To the best of our knowledge, this is the first study comparing sequence- and structure-based clustering approaches not just on specific antigen-enriched antibody sets but a more diverse repertoire.

However, limitations of the simulated repertoire should be recognized. First, a single simulated repertoire cannot fully capture the diverse characteristics of B-cell repertoires. Although our dataset contains sequences with varying degrees of mutation rates, it does not account for repertoires of differing sizes or a non-naive background for the curated antibody pairs. Future studies applying these novel tools on additional repertoires, both simulated and native, can enhance our understanding of their capabilities and limitations. Further, only kappa light chains were generated despite approximately 35% of human repertoires being comprised of lambda light chains [35]. This choice was made for simplicity considering that function is strongly driven by the antibody's heavy chain [7]. However, the research bias regarding lambda light chains has been highlighted [35], underscoring the potential value of incorporating them in future work.

SAAB+ provided one of the first attempt to use structural information to group antibodies. As the tool is based on homology modeling, it can only assign clusters for antibodies with adequate templates available in the PDB. With the recent strides in general protein structure

prediction [17] and its customization to antibody structure prediction [20], the advantage of *ab initio* structure prediction over homology modeling is compelling. The development of SPACE2, a clustering approach using the predicted antibody structures, was thus a logical next step in incorporating structural information for the analysis of antibody data.

We could confirm several findings regarding structure-based clustering approaches. Namely, SAAB+ and SPACE2 grouped the antibodies into clusters of larger size. This could suggest that more functional convergence was detected compared to conventional clustering by clonotyping. However, as annotations for the simulated repertoire are missing, no definitive evidence of functional convergence is available. (Table 2). We have shown that utilizing structural information does improve the clustering capabilities of SPACE2 and that combining CDR length matching with this property with the CDR length criterion is more efficient than combination with a sequence-based properties. Additionally, both structure-based methods were able to group some antibody pairs of low sequence similarity together (Fig 2B). This finding confirms the prospect that structural information can be used to group sequence- and genetically dissimilar but functionally converged antibodies. Further investigation is needed, however, to gain a better understanding of which specific types of functionally converged antibodies these approaches can identify and which remain undetectable. For this purpose, larger datasets of functionally converged antibodies are required.

In this study, important limitations of the currently available structure-based clustering methods already became evident and are vital to contemplate when applying these tools to analyze and select antibodies. While covering the majority of data in this study, SAAB+ is not guaranteed to assign every sequence to a cluster because of the intrinsic restrictions of homology modeling and the need of an existing template. This issue is especially prevalent for the CDRH3 region [36]. Thus, SAAB+ will not provide annotations for antibodies with unique CDRH3 sequences not covered in the PDB.

The constraint of SPACE2 for same length CDR regions raises important considerations. Our results challenge the assumption that functionally similar antibodies share identical CDR lengths [24], calling into question if this criterion should be applied. The existence of insertions and deletions within the CDR regions, which induces changes in sequence length, has been observed consistently [37,38]. While point mutations are accounting for the majority of mutations in antibodies, the rate of insertions and deletions is not insignificant [37]. The finding of differing CDRH3 length has implications for clonotyping as well. As we found functionally similar antibody pairs with differing CDRH3 length, albeit none with a CDRH3 amino acid sequence identity above 80%, we suggest to not apply this criterion when clustering antibodies based on sequence. Additionally, requiring identical J genes to assign antibodies to the same cluster should be deliberated further considering that the short length and the high similarity between J genes favor wrong gene assignments. This could lead to an inflated clonotype diversity within a repertoire.

Another source of error for structure-based clustering could stem from inconsistencies in the predicted antibody structure models. Correct modeling of the CDR regions is still challenging as these often take on unique and dynamic loop conformations [21,36,39]. Furthermore, *ab initio* antibody structure predictors produce antibody chain models without a binding partner, while an antibody's function is carried out by interacting with its antigen. The potential conformational changes upon binding, especially within the CDR loops, cannot be considered within the prediction models [40].

Before any approach is chosen to cluster repertoire data, researchers should first consider which question they wish to answer. Specifically, one might wish to identify antibodies of the same origin that are still similar enough within the CDRH3 region to likely have the same function. In this case, clonotyping will provide the most suitable grouping and will allow

further downstream analysis providing insights into the immune response, e.g., clonal expansion events. Note that the accurate identification of clonal families is an active research field with many approaches having been suggested in recent years that maximize accurate inference of antibodies with the same genetic origin [41,42].

On the other hand, if the aim is to identify antibodies that likely bind the same epitope as a candidate of interest, structure-based clustering methods such as SPACE2 might be useful, especially if the dataset contains only a small number of insertion and deletion events. A combination of methods should be contemplated, as recently suggested by Chomicz et al. [27], as the sets of identified antibody pairs in this study similarly did not completely overlap with each other. Additionally, we showed that combining the CDR length matching with structural information was more efficient than adding sequence-based information (S2 Fig), suggesting that antibody sequence and structure contain complementary insights. The high precision of all included methods, even when using less stringent cutoffs for clustering, highlight the benefit that could be gained from combining results to increase sensitivity without a strong risk for erroneous clustering of functionally dissimilarity antibodies.

This work shows that more work is required before structure-based clustering can fully deliver on its promise to detect all functionally converged antibodies. However, the developments of recent years have led to immense progress in important areas. Most importantly, annotation of full repertoires with their predicted structure is now possible [20] and more innovations are likely to follow soon.

## Methods

### Antibody pair selection

The Immune Epitope Database (IEDB) [29] (http://www.iedb.org/) collects and describes epitopes in a standardized manner. All entries of discontinuous epitopes with at least one reported positive B-cell assay and an associated 3D structure were downloaded from the IEDB on 15 February 2024. Filtering for epitopes with a known structure in the Protein Data Bank (PDB) [31] (http://www.rcsb.org/) was required as only in these cases relevant information on the binding antibody, e.g., its amino acid sequence, is also available.

The IEDB dataset was searched for antibodies with a highly similar function, i.e., antibodies that bind a largely overlapping epitope on the same protein antigen. The 50 most well-annotated antigens were selected to compare reported epitopes in a pairwise manner. Two antibodies were defined as functionally similar if the Jaccard index of their respective epitope residues is ≥0.75. The Jaccard index is a commonly used measure of similarity of two sets defined as the intersection divided by the union:

$$J(E_1, E_2) = \frac{E_1 \cap E_2}{E_1 \cup E_2} \tag{1}$$

where $E_1$ and $E_2$ describe the set of residues of the two epitopes to be compared.

The set of antibody pairs that match the epitope similarity criterion were further filtered using information of the antibody-antigen-complex PDB structure. For this purpose, the PDB for each antibody associated with an epitope reported within the IEDB was cross-referenced with the structural antibody database SAbDab [30]. Further filtering excluded antibodies of non-human origin, without definitive heavy, light and antigen chain assignments, single-chain variable fragments, as well as PDB structures solved by methods other than X-ray crystallography.

For each remaining antibody the amino acid sequences of the heavy and light chains were retrieved. The sequences were numbered using the IMGT numbering scheme and the human germline gene reference of ANARCI to identify the CDR3 regions [43]. Antibody pairs with identical receptor sequence or identical CDRH3 regions were removed to limit redundancy in the dataset. The two CDRH3 regions were compared to identify which antibody pairs show high and low sequence similarity within the CDRH3 region. The Levenshtein distance calculates the minimum number of substitutions, insertions and deletions to align two sequences [44]. The normalized Levenshtein distance was derived from the absolute Levenshtein distance as follows:

$$L_{normalized}(C_1, C_2) = 1 - \frac{L_{absolute}}{len(C_1) + len(C_2))} \tag{2}$$

where $C_1$ and $C_2$ describe the compared sequences and $L_{absolute}$ the Levenshtein distance.

## Nucleotide sequence backtranslation

To introduce the final set of antibodies of the curated antibody pair dataset into a simulated repertoire sequencing dataset, the amino acid sequences had to be backtranslated to nucleotide sequences. This process is not straightforward as translation from amino acid to nucleotide codon is ambiguous and antibody chains can originate from multiple genes. Importantly, backtranslation should not introduce any systematic bias into the antibody pair sequences, e.g., within the V/J gene assignment, to allow a fair comparison of the clustering approaches later on.

First, each heavy and light chain was compared to the human AIRR-C IG reference sets to find the most similar reference sequence for both V and J gene respectively [45]. Alignment and scoring was done using the pairwise2 module of Biopython [46]. For each mismatch between the reference gene sequence and the actual antibody sequence at the amino acid level, the codon triplet requiring the least number of single nucleotide substitutions was inserted within the nucleotide sequence of the antibody. If multiple codons had an identical number of required substitutions, information on substitution likelihoods within immunoglobulins was utilized, to choose the most likely nucleotide codon for the antibody sequence [47]. If more than one substitution within one codon was necessary, all mutation orders were compared and the most likely selected. If several reference genes had the same similarity to the antibody sequence, the reference gene requiring the least number of nucleotide substitutions was chosen. If two reference genes had the same similarity at the nucleotide level as well, the reference with the more likely substitutions was selected.

Note that Yaari et al. [47] provided the substitution likelihoods for each possible fivemer within an immunoglobulin nucleotide sequence. To simplify the backtranslation, we aggregated the fivemer substitution likelihoods to trimer substitution likelihoods to easily identify the most likely codon triplet for substitution. For instance, the substitution likelihoods for the central nucleotide position of fivemers 'AAAAA', 'AAAAC', 'AAAAG', 'AAAAT', 'CAAAA', 'CAAAC', 'CAAAG', 'CAAAT', 'GAAAA', 'GAAAC', 'GAAAG', 'GAAAT', 'TAAAA', 'TAAAC', 'TAAAG', 'TAAAT' are aggregated as trimer 'AAA' by averaging all likelihoods for each possible nucleotide substitution respectively. The process was repeated for start and end nucleotide in each trimer. We confirmed that the averaged substitution likelihoods do not display a high standard deviation (S3 Fig).

## Somatic hypermutation rate calculation

To minimize bias in the data, it is important that the backtranslated sequences of the antibody pair dataset do not contain significantly more or less somatic hypermutation (SHM) events than the simulated repertoire. Thus, the SHM rates of the curated antibodies were calculated at the nucleotide level. We corrected the SHM rate to account for the circumstance that mutations in the CDR3 region cannot be considered because of a missing aligned reference for this segment. As mutations are only introduced within the CDR regions of the simulated immunoglobulin sequence, this would lead to the SHM rate being underestimated without correction.

The distribution of the corrected SHM rates in the antibody pair sequences was used to define a low and a high SHM rate as well as a ratio between these. The low SHM rate was set at 5.1%, the high SHM rate at 11.9%.

## Repertoire data simulation

Repertoire data simulation was done using the R package immuneSIM [33]. Human heavy and kappa chains were generated and randomly merged into a heavy/light paired set. The SHM rate of the simulated repertoire was divided as follows: no SHM (0% rate) in 50%, low SHM rate (5.1%) in 36%, high SHM rate (11.9%) in 14% of sequences, respectively. This approach ensured that the simulated repertoire contains comparably mutated sequences as the curated antibodies. After introducing the curated antibody pair set into the simulated repertoire, the full repertoire comprised 10.490 unique sequences, a size dimension comparable to single-cell repertoire sequencing experiments [48]. The similarity of the curated and the simulated antibodies was confirmed using PCA for dimensionality reduction and visualization. The following sequence-based descriptors of the antibody heavy chains were included in the PCA: amino acid length of all CDR and framework regions, full receptor amino acid length, and number of V and J gene mutations.

## Repertoire clustering

Identification of antibodies from the simulated repertoire sequencing data, V/J gene assignment and clonotyping of the repertoire was performed using the IGX platform [34] apps IGX-Profile and IGX-Cluster. Clustering is conducted using agglomerative hierarchical clustering using single linkage and Levenshtein distance. IGX-Cluster merges clonotypes based on a CDRH3 amino acid sequence similarity of ≥ 80% as well as matching V/J gene assignments. Additional clonotyping settings that were tested include: 1) matching V gene assignment and ≥ 80% CDRH3 sequence identity, 2) matching V/J gene assignment and ≥ 70% CDRH3 sequence identity, 3) matching V gene assignment and ≥ 70% CDRH3 sequence identity, 4) no gene matching and ≥ 80% CDRH3 sequence identity. Unless stated otherwise, sequence identity is calculated at the amino acid level.

Clustering by SAAB+ [25] was performed with default setting.

Clustering by SPACE2 [24] required the structure modeling of all repertoire sequences beforehand. Heavy and light chain structures for all antibodies were predicted with ImmuneBuilder [20]. SPACE2 was used to cluster these models using the default RMSD cutoff of 1.25Å and the agglomerative clustering algorithm. Clustering was repeated with varying RMSD thresholds between 1 and 2.5Å.

### Performance evaluation and comparison

Multiple metrics were compared for each clustering approach including the coverage, i.e., number of clustered antibodies, number of clusters, and mean cluster size distribution. Number and mean cluster size of multiple-occupancy clusters, i.e., clusters that contain two or more distinct antibodies [24]), was also included. The antibody pair dataset was analyzed further as it is the subset of the repertoire for which antibody binding (i.e., epitope) information is available. For each method, the number of correctly clustered antibody pairs, and the number of correctly clustered low-sequence similarity antibody pairs, i.e., with a CDRH3 sequence identity <0.8, were calculated. The average epitope overlap and CDRH3 sequence identity of correctly clustered antibody pairs per method were determined.

A random clustering rate was established for each method and each tested setting. To calculate this rate, the cluster size distribution for each clustering experiment was gathered. The repertoire antibodies were then randomly assigned to clusters of the respective sizes to infer how likely random assignment to the same cluster is for the antibody pairs. The number of randomly clustered antibody pairs was divided by the number of antibody pairs to calculate the random clustering rate. This process was repeated 1000 times and the rate averaged.

## Supporting information

**S1 Fig. PCA of full simulated repertoire using sequence descriptors.** A PCA was fitted using sequence descriptors of the simulated repertoire and the antibody pair set. The first and second principal component are shown. The annotated (pink) and simulated (beige) antibodies overlap strongly, indicating that these antibody sets are not consistently different from each other.
(TIF)

**S2 Fig. Random clustering rates and sensitivity of clustering strategies using the CDR length criterion.** A: The random clustering rate is significantly higher when clustering is solely based on CDR length, compared to approaches that incorporate structural or CDRH3 sequence information as well. B: Combination of CDR length and CDRH3 sequence identity leads to the lowest sensitivity. Relaxing the RMSD cutoff increases sensitivity until all identical CDR lengths antibody pairs are correctly identified.
(TIF)

**S3 Fig. Aggregated substitution likelihoods.** Observed and inferred substitution likelihoods of single nucleotides within immune receptor sequences have been used for backtranslation of amino acid to nucleotide sequences of the curated antibodies. The substitution likelihoods were provided for fivemers. To simplify backtranslation these fivemer substitution likelihoods were aggregated to trimer substitution likelihoods. The aggregated likelihoods for each substitution including standard deviations are shown for the start (A), center (B), and end (C) nucleotide of each possible trimer.
(TIF)

**S1 Table. Annotated antibody pair dataset.** Extended information on the final set of antibody pairs including associated IEDB and PDB IDs, information on the antigen and epitope residues, Jaccard index of the epitope overlap, and Levenshtein distance of the CDRH3 region.
(XLSX)

## Acknowledgments

We would like to acknowledge BAZIS, the Supercomputing cluster of the Vrije Universiteit Amsterdam.

## Author contributions

**Conceptualization:** Katharina Waury, Henk-Jan van den Ham.

**Data curation:** Katharina Waury.

**Formal analysis:** Katharina Waury.

**Funding acquisition:** Sanne Abeln.

**Investigation:** Katharina Waury.

**Methodology:** Katharina Waury, Stefan Lelieveld, Henk-Jan van den Ham.

**Project administration:** Sanne Abeln, Henk-Jan van den Ham.

**Resources:** Sanne Abeln, Henk-Jan van den Ham.

**Software:** Katharina Waury, Stefan Lelieveld.

**Supervision:** Sanne Abeln, Henk-Jan van den Ham.

**Visualization:** Katharina Waury.

**Writing – original draft:** Katharina Waury.

**Writing – review & editing:** Katharina Waury, Stefan Lelieveld, Sanne Abeln, Henk-Jan van den Ham.

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
