## [Decision Letter · Decision Letter 0]

12 Nov 2024

PCOMPBIOL-D-24-01065Comparison of sequence- and structure-based antibody clustering approaches on simulated repertoire sequencing dataPLOS Computational Biology Dear Dr. van den Ham, Thank you for submitting your manuscript to PLOS Computational Biology. After careful consideration, we feel that it has merit but does not fully meet PLOS Computational Biology's publication criteria as it currently stands. Therefore, we invite you to submit a revised version of the manuscript that addresses the points raised during the review process. Please submit your revised manuscript within 60 days Jan 12 2025 11:59PM. If you will need more time than this to complete your revisions, please reply to this message or contact the journal office at ploscompbiol@plos.org. Please include the following items when submitting your revised manuscript: * A rebuttal letter that responds to each point raised by the editor and reviewer(s). You should upload this letter as a separate file labeled 'Response to Reviewers'. This file does not need to include responses to formatting updates and technical items listed in the 'Journal Requirements' section below.* A marked-up copy of your manuscript that highlights changes made to the original version. You should upload this as a separate file labeled 'Revised Manuscript with Track Changes'.* An unmarked version of your revised paper without tracked changes. You should upload this as a separate file labeled 'Manuscript'. If you would like to make changes to your financial disclosure, competing interests statement, or data availability statement, please make these updates within the submission form at the time of resubmission. Guidelines for resubmitting your figure files are available below the reviewer comments at the end of this letter. We look forward to receiving your revised manuscript. Kind regards, Claude Loverdo, Ph.D.Academic EditorPLOS Computational Biology Amber SmithSection EditorPLOS Computational Biology Feilim Mac GabhannEditor-in-ChiefPLOS Computational Biology Jason PapinEditor-in-ChiefPLOS Computational Biology  **Journal Requirements:** **Additional Editor Comments (if provided):** While all the reviewers had positive things to say about the manuscript, reviewer 3 raised substantial points that should be addressed.**Reviewers' comments:** Reviewer's Responses to Questions

**Comments to the Authors:**

Reviewer #1: In this paper Waury et al. investigate the utility of antibody structural clustering methods, specifically SAAB+ and SPACE2, for their ability to group sequences that bind to the same epitope on a given antigen amidst a diverse simulated repertoire. They compare these two structural clustering methods to the traditional method of ‘clonotyping’ which relies on the CDRH3 sequence identity and V/J genes to group similar sequences. Structural methods instead group by shape of the paratope region and therefore can identify relationships which may not be evident from sequence alone. SAAB+ relies on homology modelling whereas SPACE2 clusters length matched CDRs. The ability of these methods to group pairs of antibodies which bind the same epitope amidst a large artificial repertoire was analysed.

No method was able to correctly group most of epitope-specific pairs antibody pairs. While clonotyping outperformed both structural methods, the authors demonstrated that structural clustering did group some pairs with highly dissimilar CDRH3 sequences. The limitations of each method are also assessed in detail. Of particular interest were observations the CDR length matching applied by SPACE2 is very restrictive, as well as the ability to increase cluster size (through lower stringency) without overly compromising specificity.

Overall, this paper is well thought out, figures are nicely presented, and the text is clearly written. I have only minor comments.

Minor comments

1. Line 199-201: Although the authors have tried to explain how pairs can be found in the clonotyping method that exceed the identity threshold, it is still not fully clear (assume they mean that within a cluster the overall distance might be greater, but distance from the cluster centroid is still below the threshold). Please spell this out exactly.

2. Line 435-437: In generating the artificial dataset. Does it matter that the authors have not used lambda chains and only focused on kappa? Please explain why?

3. Figure 2: Please state the statistical test used in the legend.

4. Line 116: Typo – should say clusters (plural).

Reviewer #2: The authors curated a dataset of antibody pairs confirmed to bind to the same epitope; they then placed these into a simulated naïve repertoire, clustered the simulated repertoire via 3 algorithms and looked at the sensitivity and specificity of these methods in the context of a naïve human background. In contrast to previous findings, they found that the sequence-based method had the highest sensitivity, and was able to identify a larger number of sequence-dissimilar/same-epitope pairs. The methodology is more precise and realistic than previous benchmarks, the claims are well-substantiated, and the dataset could be widely-used.

Minor

- Line 26 – DOI in text.

- Line 93 - Capitalize Immune Epitope Database

- Please cite the original SPACE paper.

- Line 58-59 - SPACE2 can work with any numbered antibody structure (predicted with ImmuneBuilder or otherwise), so this is not entirely accurate (although indeed this is recommended).

- Line 197 – Is this not just a consequence of hierarchical clustering?

- Figure 4 caption mentions specificity but I can’t see this plotted anywhere (presumably because it is perfect at all thresholds)? It says “specificity stays high” – I would just say “specificity remains 100%”.

- I disagree with the usage of “functional clustering” to refer to non-validated clusterings in the simulated data - there is no evidence of shared function in these large clusters.

- It is not clear to me whether having any RMSD cut-off at all improves specificity, or whether most of SPACE2’s predictive power is in CDR length matching. At the highest cut-off there is still perfect specificity. Please could you report sensitivity and specificity for the CDR length matching alone. Further to this - I haven’t seen a simple CDR sequence identity clustering be compared to SPACE2. Please could you calculate the sensitivity+specificity using sequence identity instead of RMSD in the SPACE2 clustering algorithm- a threshold could be selected to produce a similar number of clusters to RMSD clustering or you could consider a reasonable range such as 70% and above. This would be stronger evidence that SPACE2 works because of structure specifically.

Reviewer #3: The manuscript by Waury et al. compares two recent approaches to detect functionally related antibodies from B cell repertoire data, SSAB+ and SPACE2 (refs. 22 and 23). The benchmark highlights several interesting features of these methods and points out significant limitations in applying structure-based algorithms to real repertoires.

A comparison of existing methods for detecting convergent evolution is important to researchers analyzing immune repertoires. However, the benchmark presented in this study is limited in scope as it only considers a single synthetic repertoire with a trivial clonal structure. Some of the conclusions, in particular regarding the comparison of structure-based methods with clonotyping, are poorly supported by evidence presented in the manuscript.

I think two factors significantly diminish the methodological insight of this paper and its potential impact.

1. A single synthetic dataset was used for comparison and its parameters don't reflect the potential difficulties in detecting convergent evolution. These include varying degrees of mutation, repertoire size (depth of sampling), and, seemingly most important for the task at hand, a nontrivial clonotype structure. Antibody repertoires come in multiple shapes and sizes, and a meaningful comparison of clustering approaches must take that into account.

2. The comparison with clonotyping is done with a relatively old method, which suffers from low accuracy, as suggested by more recent studies (see the following references and comparisons to other methods there: Ralph and Matsen, 2016 and 2022; Nouri and Kleinstein, 2020, Lindenbaum et al., 2020; Spisak et al., 2024), as well as by the results presented here (IGX-Cluster merges sequences that are not clonally related and have >40% divergence in CDRH3, in larger datasets this leads to positive predictive value close to zero).

Links to references:

Ralph and Matsen, 2016, doi.org/10.1371/journal.pcbi.1005086

Nouri and Kleinstein, 2020, doi.org/10.1371/journal.pcbi.1007977

Lindenbaum et al., 2020, doi.org/10.1093/nar/gkaa1160

Ralph and Matsen, 2022, doi.org/10.1371/journal.pcbi.1010723

Spisak et al., 2024, doi.org/10.7554/eLife.86181

The paper is appropriately structured. Data and code are readily available and the methodology is clearly described in the manuscript. The clarity of the presentation could be improved by avoiding the use of jargon. A few concrete suggestions and comments are listed below.

1. Abstract: I'm afraid the phrase "multiple-occupancy clusters" will be confusing to the reader, and probably shouldn't be introduced in the abstract. I find the formulation in "author's summary" easier to follow.

2. The reference in line 26 is not formatted.

3. In line 27: "them" suggests "some of them".

4. In line 84: this sentence is hard to parse

5. In line 95: the 75% threshold is not justified.

6. When discussing sequence identity, it should be stated whether this refers to amino acid or nucleotide sequence.

7. In line 133: suggest avoiding "spiked"

8. In line 149: The sentence "Antibodies (...)" is not grammatical.

9. In line 167. One cannot infer that any of the methods is highly specific from the observation that no two curated antibodies were incorrectly assigned to the same cluster. More generally, in discussing the performance of the methods, it's not specificity but positive predictive value or precision, that's 1. a relevant measure of accuracy and 2. difficult to achieve in clonotyping.

10. In line 184: suggests rather "identified by"?

**Have the authors made all data and (if applicable) computational code underlying the findings in their manuscript fully available?**

Reviewer #1: Yes

Reviewer #2: Yes

Reviewer #3: Yes

PLOS authors have the option to publish the peer review history of their article (what does this mean?). If published, this will include your full peer review and any attached files.

Reviewer #1: **Yes: **Charlotte Deane

Reviewer #2: No

Reviewer #3: No

 **Figure resubmission:**While revising your submission, please upload your figure files to the Preflight Analysis and Conversion Engine (PACE) digital diagnostic tool, https://pacev2.apexcovantage.com/. PACE helps ensure that figures meet PLOS requirements. To use PACE, you must first register as a user. Registration is free. Then, login and navigate to the UPLOAD tab, where you will find detailed instructions on how to use the tool. If you encounter any issues or have any questions when using PACE, please email PLOS at figures@plos.org. Please note that Supporting Information files do not need this step. If there are other versions of figure files still present in your submission file inventory at resubmission, please replace them with the PACE-processed versions. 
---

## [Decision Letter · Decision Letter 1]

17 Apr 2025

Dear van den Ham,

We are pleased to inform you that your manuscript 'Comparison of sequence- and structure-based antibody clustering approaches on simulated repertoire sequencing data' has been provisionally accepted for publication in PLOS Computational Biology.

Best regards,

Claude Loverdo, Ph.D.

Academic Editor

PLOS Computational Biology

Amber Smith

Section Editor

PLOS Computational Biology

Reviewer's Responses to Questions

**Comments to the Authors:**

Reviewer #1: The authors have addressed all of the points in their response.

Reviewer #2: All of my comments were addressed well. I congratulate the authors for their valuable contribution to this field.

Reviewer #3: The response by the authors is lengthy but not consequential; the manuscript hasn't gone through a major revision. As one example, my comment about using PPV rather than specificity to quantify the methods' accuracy has led to rephrasing of one or two sentences but not a change in data analysis. The actual statistic used in the figures is "random clustering rate," and (as far as I can from its confusing definition on page 16) it's neither specificity nor PPV (and certainly not both).

The main points of my critique have not been addressed. The analysis relies on a single synthetic dataset. No evidence is presented on how closely it matches realistic repertoires beyond the PCA plot in the supplementary figure. This plot is uninformative because (1) the simulated dataset was supposed to resemble a typical repertoire and not the curated dataset of antibody pairs, and (2) it's not clear what characteristics of the datasets are picked up by the two principal components.

As a consequence, a number of claims made by the authors remain poorly supported.

**Have the authors made all data and (if applicable) computational code underlying the findings in their manuscript fully available?**

Reviewer #1: Yes

Reviewer #2: Yes

Reviewer #3: None

PLOS authors have the option to publish the peer review history of their article (what does this mean?). If published, this will include your full peer review and any attached files.

Reviewer #1: **Yes: **Charlotte Deane

Reviewer #2: **Yes: **Eve Richardson

Reviewer #3: No

---

## [Editor Report · Acceptance letter]

PCOMPBIOL-D-24-01065R1

Comparison of sequence- and structure-based antibody clustering approaches on simulated repertoire sequencing data

Dear Dr van den Ham,

I am pleased to inform you that your manuscript has been formally accepted for publication in PLOS Computational Biology. Your manuscript is now with our production department and you will be notified of the publication date in due course.

With kind regards,

Judit Kozma
